# Joint Optimal Transport and Embedding for Network Alignment

## Abstract

Network alignment, which aims to find node correspondence across different networks, is the cornerstone of various downstream multi-network and Web mining tasks. Most of the embedding-based methods indirectly model cross-network node relationships by contrasting positive and negative node pairs sampled from hand-crafted strategies, which are vulnerable to graph noises and leads to potential misalignment of nodes. Another line of works based on the optimal transport (OT) theory directly model cross-network node relationships and generate noise-reduced alignments. However, OT methods heavily rely on fixed, pre-defined cost functions that prohibit end-to-end training and are hard to generalize. In this paper, we aim to unify the embedding and OT-based methods in a mutually beneficial manner and propose a joint optimal transport and embedding framework for network alignment named JOENA. For one thing (*OT for embedding*), through a simple yet effective transformation, the noise-reduced OT mapping serves as an adaptive sampling strategy directly modeling all cross-network node pairs for robust embedding learning. For another (*embedding for OT*), on top of the learned node embeddings, the OT cost can be gradually trained along the learning process in an end-to-end fashion, which further enhances the alignment quality. With a unified objective, the mutual benefits of both methods can be achieved by an alternating optimization schema with guaranteed convergence. Extensive experiments on real-world networks validate the effectiveness and scalability of JOENA, achieving up to 16% improvement in MRR and 20× speedup compared with the state-of-the-art alignment methods.

## Keywords

Network Alignment, Optimal Transport, Network Embedding

## 1 Introduction

In the era of big data and AI, multi-sourced networks[1] appear in a wealth of high-impact applications, ranging from social network analysis [2, 36], financial fraud detection [41, 42], to knowledge graphs [29, 33]. Network alignment, the process of identifying node associations across different networks, is the key steppingstone behind many downstream multi-network and Web mining tasks. For example, by linking users across different social network platforms, we can integrate user actions from multi-sourced sites to achieve more informed and personalized recommendation [2, 6, 36]. Aligning suspects from different transaction networks helps identify financial fraud [41, 42]. Entity alignment between complementary incomplete knowledge graphs, such as Wikipedia and WorkNet, helps construct a unified knowledge base [4, 29, 33].

Many existing methods approach the network alignment problem by learning low-dimensional node embeddings in a unified space across two networks. Essentially, these methods first adopt different sampling strategies to sample positive and negative node

---

[1]In this paper, we use the terms 'network' and 'graph' interchangeably.

pairs, and then utilize a ranking loss, where positive node pairs (e.g., anchor nodes) are pulled together, while negative node pairs (e.g., sampled dissimilar nodes) are pushed far apart in the embedding space, to model cross-network node relationships [6, 14, 34, 41]. For example, as shown in Figure 1, the relationship between anchor node pair $(a_1, a_2)$ is *directly* modeled by minimizing their distance $d(a_1, a_2)$ in the embedding space, while the relationship between $(b_1, b_2)$ is depicted via an *indirect* modeling path $d(b_1, a_1) + d(a_1, a_2) + d(a_2, b_2)$ [34].

Promising as it might be, the indirect modeling adopted by embedding-based methods inevitably bear an approximation error between the path $d(b_1, a_1) + d(a_1, a_2) + d(a_2, b_2)$ and the exact cross-network node relationship $d(b_1, b_2)$, resulting in performance degradation. Besides, embedding-based methods largely depend on the quality of node pairs sampled by hand-crafted sampling strategies such as random walk-based [32], degree-based [6, 14] and similarity-based [34, 41] strategies. However, such hand-crafted strategies often suffer from high vulnerability to graph noises (e.g., structural and attribute noise), further exacerbating the detrimental effect of indirect modeling. For example, as shown in Figure 1, when modeling the relationship between $(b_1, b_2)$ with a missing edge, $(b_1, a_1)$ will be misidentified as an intra-network negative pair by the random walk-based strategy, and the indirect modeling $d(b_1, a_1) + d(a_1, a_2) + d(a_2, b_2)$ will be enlarged as the ranking loss tends to increase $d(b_1, a_1)$, hence failing to align $b_1$ and $b_2$. Similarly, due to attribute noise on $d_1$, the false negative intra-network node pair $(a_1, d_1)$ sampled by the similarity-based strategy will push the to-be-aligned node pair $(d_1, d_2)$ far apart. Besides, as the amount of indirectly modeled non-anchor node pairs (grey squares in Figure 1) is significantly greater than directly modeled anchor node pairs (colored squares in Figure 1), the effect of false negative pairs will be further exacerbated.

Another line of works utilize the optimal transport (OT) theory for network alignment. By transforming graphs as distributions over the node set, network alignment can be formulated as a distributional matching problem based on a transport cost function measuring cross-network node distances. Thanks to the marginal constraints in OT [18], OT-based method generates noise-reduced alignment with soft one-to-one matching [38]. However, the effectiveness of most, if not all, of the existing OT-based methods largely depend on pre-defined cost functions, focusing on specific graph structure [10, 15, 17, 21] or node attributes [3, 38], leading to relatively poor generalization capability. Though efforts have been made to combine both methods by adopting the OT objective to supervise embedding learning [3, 5, 25, 30, 31], we theoretically reveal that directly applying the OT objective for embedding learning cause *embedding collapse* where all nodes are mapped to an identical point in the embedding space, hence dramatically degrading the discriminating power.

In light of the pros and cons of embedding-based and OT-based methods, we seek to explore the complementary roles of two categories of methods to fully realize their mutual benefits. Specifically,

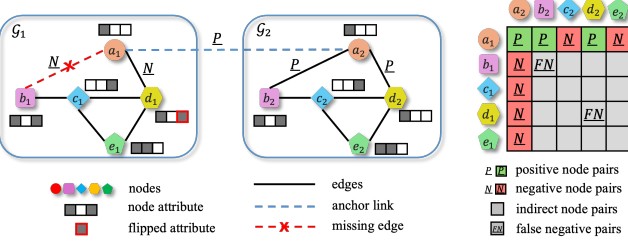

**Figure 1: An example of embedding-based methods with hand-crafted sampling strategies. Due to edge noise, $(a_1, b_1)$ is identified as a false negative intra-network pair, pushing $(b_1, b_2)$ that should be aligned far apart. Likewise, $(d_1, d_2)$ fails to align due to attribute noise on $d_1$. Best viewed in color.**

we first demonstrate their close intrinsic relationships: the OT objective can be neatly transformed into a multi-level ranking loss with a weighted sampling strategy. Based on this theoretical finding, we propose a novel unified framework named JOENA to learn node embeddings and alignments jointly in a mutually beneficial way. For one thing, to augment embedding learning with OT, the OT mapping is transformed into a cross-network sampling strategy, which not only helps avoid embedding collapse, but also enhances model robustness against graph noises thanks to the direct modeling and noise-reduced property of OT [23, 38]. For another, to augment OT with embedding learning, JOENA utilizes the learned node embeddings for a better OT cost design, which opens the door for the end-to-end training paradigm and can be adapted to different graphs without extensive parameter tuning. We have compared the proposed JOENA with the state-of-the-art network alignment methods on six different datasets, which validates the effectiveness and efficiency of our proposed model.

The main contributions of this paper are summarized as follows:

- **Theoretical Analysis.** To our best knowledge, we are the first to theoretically reveal the close relationship and mutual benefits between OT and embedding-based methods.
- **Novel Model.** We propose a novel framework JOENA to learn node embeddings and alignments jointly based on a unified objective function.
- **Extensive Experiments.** Extensive results on real-world datasets demonstrate the effectiveness and scalability of JOENA, with up to 16% and 6% outperformance in MRR on plain and attributed networks, and up to 20× speed-up in inference time.

The rest of the paper is organized as follows. Section 2 defines the network alignment problem and introduces the preliminaries. Section 3 presents the proposed JOENA and relevant analysis. Section 4 shows the experimental results. Related works and conclusions are given in Section 5 and Section 6, respectively.

## 2 Preliminaries

Table 1 summarizes the main symbols used throughout the paper. We use bold uppercase letters for matrices (e.g., $\mathbf{A}$), bold lowercase letters for vectors (e.g., $\mathbf{s}$), and lowercase letters for scalars (e.g., $\alpha$). The transpose of $\mathbf{A}$ is denoted by the superscript $\top$ (e.g., $\mathbf{A}^\top$).

**Table 1: Symbols and Notations.**

| Symbol | Definition |
|---|---|
| $\mathcal{G}_1, \mathcal{G}_2$ | input networks |
| $\mathcal{V}_1, \mathcal{V}_2$ | node sets of $\mathcal{G}_1$ and $\mathcal{G}_2$ |
| $\mathcal{E}_1, \mathcal{E}_2$ | edge sets of $\mathcal{G}_1$ and $\mathcal{G}_2$ |
| $\mathbf{A}_1, \mathbf{A}_2$ | adjacency matrices of $\mathcal{G}_1$ and $\mathcal{G}_2$ |
| $\mathbf{X}_1, \mathbf{X}_2$ | node attribute matrices of $\mathcal{G}_1$ and $\mathcal{G}_2$ |
| $\boldsymbol{\mu}_1, \boldsymbol{\mu}_2$ | probability measures |
| $n_i, m_i$ | number of nodes/edges in $\mathcal{G}_i$ |
| $\mathcal{L}$ | the set of anchor node pairs |
| $\mathbf{I}, \mathbf{1}$ | an identity matrix and an all-one vector/matrix |
| $\odot$ | Hadamard product |
| $\langle \cdot, \cdot \rangle$ | inner product |
| $\Pi$ | probabilistic coupling |
| $[\cdot \| \cdot]$ | horizontal concatenation of vectors |

An attributed network with $n$ nodes is represented by $\mathcal{G} = (\mathbf{A}, \mathbf{X})$ where $\mathbf{A} \in \mathbb{R}^{n \times n}, \mathbf{X} \in \mathbb{R}^{n \times d}$ denote the adjacency matrix and node attribute matrix, respectively. We use $\mathcal{V}$ and $\mathcal{E}$ to denote the node and edge set of a graph, respectively. The semi-supervised attributed network alignment problem can be defined as follows:

**Definition 1.** *Semi-supervised Attributed Network Alignment.*
**Given:** *(1) two networks $\mathcal{G}_1 = (\mathbf{A}_1, \mathbf{X}_1)$ and $\mathcal{G}_2 = (\mathbf{A}_2, \mathbf{X}_2)$; (2) an anchor node set $\mathcal{L} = \{(x, y) | x \in \mathcal{G}_1, y \in \mathcal{G}_2\}$ indicating pre-aligned nodes pairs $(x, y)$.*
**Output:** *alignment/mapping matrix $\mathbf{S} \in \mathbb{R}^{n_1 \times n_2}$, where $\mathbf{S}(x, y)$ indicates how likely node $x \in \mathcal{G}_1$ and node $y \in \mathcal{G}_2$ are aligned.*

### 2.1 Embedding-based Network Alignment

Embedding-based methods learn node embeddings by pulling positive node pairs together while pushing negative node pairs apart in the embedding space via ranking loss functions [6, 14, 34, 41]. Specifically, given a set of anchor node pairs $\mathcal{L}$, the ranking loss can be generally formulated as [34]:

$$\mathcal{J}_{\text{rank}} = \mathcal{J}_1 + \mathcal{J}_2 + \mathcal{J}_{\text{cross}}$$

$$\text{where} \begin{cases} \mathcal{J}_1 = \sum_{x \in \mathcal{L} \cap \mathcal{G}_1} \left( d(x, x_p) - d(x, x_n) \right) \\ \mathcal{J}_2 = \sum_{y \in \mathcal{L} \cap \mathcal{G}_2} \left( d(y, y_p) - d(y, y_n) \right), \\ \mathcal{J}_{\text{cross}} = \sum_{(x,y) \in \mathcal{L}} d(x, y) \end{cases} \quad (1)$$

where $d(x, y)$ measures the distance between two node embeddings (e.g., $L_1$ norm), $x_p/y_p$ denotes the positive node w.r.t. $x/y$, and $x_n/y_n$ denotes the negative node w.r.t. $x/y$. In the above equation, $\mathcal{J}_1, \mathcal{J}_2$ are intra-network loss pulling sampled positive nodes (e.g., similar/nearby nodes) together, while pushing sampled negative nodes (e.g., dissimilar/distant nodes) far part. $\mathcal{J}_{\text{cross}}$ is the cross-network loss, which aims to minimize the distance between anchor node pairs. In general, the objective in Eq. (1) indirectly models the node relationship between two non-anchor nodes $(x', y')$ via a path through the anchor node pair $(x, y)$, i.e., $((x', x), (x, y), (y, y'))$.

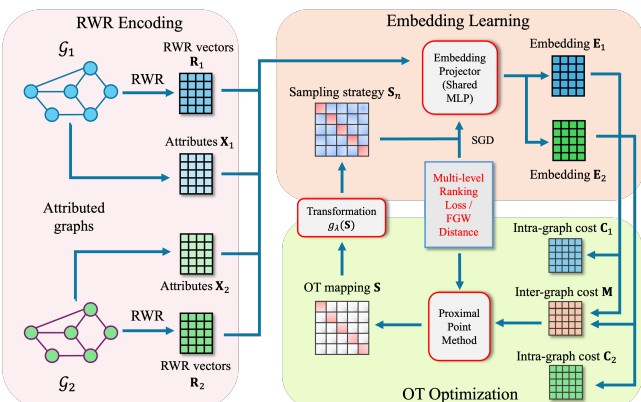

**Figure 2: An overview of JOENA, including RWR encoding, embedding learning and OT optimization. RWR encoding and raw node attributes are processed by a shared MLP projector, supervised by the ranking loss based on the OT-based sampling strategy. The OT mapping is optimized based on cost matrices derived from the learned embeddings, further transformed into a sampling strategy based on the learnable transformation $g_\lambda$.**

## 2.2 Optimal Transport

OT has recently achieved great success in graph applications, such as network alignment [25, 30, 31, 38] and graph classification [8, 19]. Following a common practice in OT-based graph applications [26], a graph can be represented as a probability measure supported on the product space of node attribute and structure, i.e., $\boldsymbol{\mu} = \sum_{i=1}^{n} \mathbf{h}(i)\delta_{\mathbf{A}(x_i),\mathbf{X}(x_i)}$, where $\mathbf{h} \in \Delta_n$ is a histogram representing the node weight and $\delta$ denotes the Dirac function. The fused Gromov-Wasserstein (FGW) distance is the sum of node pairwise distances based on node attributes and graph structure defined as [23]:

DEFINITION 2. *Fused Gromov-Wasserstein (FGW) distance.*
***Given:*** *(1) two graphs $\mathcal{G}_1 = (\mathbf{A}_1, \mathbf{X}_1), \mathcal{G}_2 = (\mathbf{A}_2, \mathbf{X}_2)$; (2) probability measures $\boldsymbol{\mu}_1, \boldsymbol{\mu}_2$ on graphs; (3) intra-network cost matrix $\mathbf{C}_1, \mathbf{C}_2$; (4) cross-network cost matrix $\mathbf{M}$.*
***Output:*** *the FGW distance between two graphs $\mathrm{FGW}_{q,\alpha}(\mathcal{G}_1, \mathcal{G}_2)$*

$$
\begin{aligned}
\min_{\mathbf{S} \in \Pi(\boldsymbol{\mu}_1, \boldsymbol{\mu}_2)} \; & (1-\alpha) \sum_{x \in \mathcal{G}_1, y \in \mathcal{G}_2} \mathbf{M}^q(x,y)\mathbf{S}(x,y) \\
& + \alpha \sum_{\substack{x,x' \in \mathcal{G}_1 \\ y,y' \in \mathcal{G}_2}} |\mathbf{C}_1(x,x') - \mathbf{C}_2(y,y')|^q \mathbf{S}(x,y)\mathbf{S}(x',y').
\end{aligned} \tag{2}
$$

The first term corresponds to the Wasserstein distance measuring cross-network node distances, and the second term is the Gromov-Wasserstein (GW) distance measuring cross-network edge distances. The hyperparameter $\alpha$ controls the trade-off between two terms, and $q$ is the order of the FGW distance, which is adopted as $q = 2$ throughout the paper. The FGW problem aims to find an OT mapping $\mathbf{S} \in \Pi(\boldsymbol{\mu}_1, \boldsymbol{\mu}_2)$ that minimizes the sum of Wasserstein and GW distances, and the resulting OT mapping matrix $\mathbf{S}$ further serves as the soft node alignment.

## 3 Methodology

In this section, we present the proposed JOENA. We first analyze the mutual benefits between embedding and OT-based methods in Section 3.1. Guided by such analysis, a unified framework named JOENA is proposed for network alignment in Section 3.2. We further present the unified model training schema in Section 3.3, followed by convergence and complexity analysis of JOENA in Section 3.4.

## 3.1 Mutual Benefits of Embedding and OT

*3.1.1 OT-Empowered Embedding Learning.* The success of ranking loss largely depends on the sampled positive and negative node pairs, i.e., $(x, x_p), (x, x_n), (y, y_p), (y, y_n)$ in Eq. (1), through which cross-network node pair relationships can be modeled. To provide a better sampling strategy, the OT mapping $\mathbf{S}$ improves the embedding learning from two aspects: *direct modeling* and *robustness*. First (*direct modeling*), while embedding-based methods model cross-network node relationships via an indirect path (see Figure 1 for an example.) sampled by hand-crafted strategies, the OT mapping directly models such cross-network relationships, identifying positive and negative node pairs more precisely. Second (*robustness*), in contrast to the noisy embedding alignment, thanks to the marginal constraints in Eq. (2), the resulting OT mapping is noise-reduced [23, 38], where each node only aligns with very few nodes. Therefore, sampling with OT-based strategy can be robust to graph noises.

*3.1.2 Embedding-Empowered OT Learning.* The success of OT-based alignment methods largely depend on the cost design, i.e. $\mathbf{C}_1$, $\mathbf{C}_2$, and $\mathbf{M}$ in Eq. (2), which is often hand-crafted in existing works. To achieve better cost design, embedding learning benefits OT learning from two aspects: *generalization* and *effectiveness*. For one thing (*generalization*), building transport cost upon learnable embeddings opens the door for end-to-end training paradigm, thus, the OT framework can be generalized to different graphs without extensive parameter tuning. For another (*effectiveness*), neural networks generate more powerful node embeddings via deep transformations, enhancing the cost design for OT optimization.

## 3.2 Model Overview

The overall framework of JOENA is given in Figure 2, which can be divided into three parts: (1) *RWR encoding* for structure learning, (2) *embedding learning* via multi-level ranking loss with OT-based sampling, (3) *OT optimization* with learnable transport cost.

Positional information plays a pivotal role in network alignment [34, 38], but most of the GNN architectures fall short in capturing such information for alignment [37]. Therefore, we explicitly encode positional information by conducting random walk with restart (RWR) [27]. By regarding a pair of anchor nodes $(x, y) \in \mathcal{L}$ as identical in the RWR embedding space, we simultaneously perform RWR w.r.t. $x \in \mathcal{G}_1$ and $y \in \mathcal{G}_2$ to construct a unified embedding space, where the RWR score vectors $\mathbf{r}_x \in \mathbb{R}^{n_1}$ and $\mathbf{r}_y \in \mathbb{R}^{n_2}$ can be obtained by [27, 34]

$$
\mathbf{r}_x = (1-\beta)\mathbf{W}_1\mathbf{r}_x + \beta\mathbf{e}_x, \; \mathbf{r}_y = (1-\beta)\mathbf{W}_2\mathbf{r}_y + \beta\mathbf{e}_y, \tag{3}
$$

where $\beta$ is the restart probability, $\mathbf{W}_i = \left(\mathbf{D}_i^{-1}\mathbf{A_i}\right)^{\top}$ is the transpose of the row-normalized adjacency matrix, $\mathbf{D}_i$ is the diagonal degree matrix of $\mathcal{G}_i$, and $\mathbf{e}_x, \mathbf{e}_y$ are one-hot encoding vectors with $\mathbf{e}_x(x) =$

1 and $\mathbf{e}_y(y) = 1$, respectively. The concatenation of RWR vectors w.r.t. different anchor nodes $\mathbf{R}_i \in \mathbb{R}^{n_i \times |\mathcal{L}|}$, together with node attribute matrices $\mathbf{X}_i$, i.e., $[\mathbf{R}_i \| \mathbf{X}_i]$, serve as the input for embedding learning.

To learn powerful node embeddings, we train a shared two-layer multi-layer perceptron (MLP) with residual connections $f_\theta$ via a multi-level ranking loss. To address the limitations of hand-crafted sampling strategies, we apply a simple yet effective transformation $g_\lambda$ on the OT mapping $\mathbf{S}$ to obtain an adaptive sampling strategy $g_\lambda(\mathbf{S})$. Then, the sampled node and edge pairs based on $g_\lambda(\mathbf{S})$ are utilized for learning output embeddings $\mathbf{E}_1$ and $\mathbf{E}_2$, supervised by the multi-level ranking loss.

To improve OT optimization, we construct the cross-network cost matrix $\mathbf{M}$ and intra-network cost matrices $\mathbf{C}_1, \mathbf{C}_2$ based on output embeddings $\mathbf{E}_1$ and $\mathbf{E}_2$ of the MLP as follows

$$\mathbf{M} = e^{-\mathbf{E}_1 \mathbf{E}_2}, \ \mathbf{C}_i = e^{-\mathbf{E}_i \mathbf{E}_i} \odot \mathbf{A}_i, \tag{4}$$

where $\mathbf{M}(x, y)$ denotes the cross-network node distance between $x \in \mathcal{G}_1, y \in \mathcal{G}_2$, and $\mathbf{C}_i(a, b)$ denotes the intra-network node distance between $a, b \in \mathcal{G}_i{}^2$. Afterwards, the FGW distance in Eq. (2) can be efficiently solved via the proximal point method [30, 38], whose output OT mapping $\mathbf{S}$ indicates the node alignment between two graphs.

For model training, we propose an objective function which, as we will show in the next subsection, unifies OT optimization and embedding learning as follows:

$$\min_{\mathbf{S} \in \Pi(\boldsymbol{\mu}_1, \boldsymbol{\mu}_2), \lambda, \theta} \mathcal{J}(\mathbf{S}, \lambda, \theta) = (1 - \alpha) \underbrace{\sum_{x \in \mathcal{G}_1, y \in \mathcal{G}_2} \mathbf{M}(x, y; \theta) \mathbf{S}_n(x, y; \lambda)}_{\text{Wasserstein/node-level loss}}$$

$$+ \alpha \underbrace{\sum_{\substack{x, x' \in \mathcal{G}_1 \\ y, y' \in \mathcal{G}_2}} |\mathbf{C}_1(x, x'; \theta) - \mathbf{C}_2(y, y'; \theta)|^2 \mathbf{S}_n(x, y; \lambda) \mathbf{S}_n(x', y'; \lambda)}_{\text{GW/edge-level loss}}, \tag{5}$$

where $\mathbf{S}$ is the OT mapping, $\theta$ is the set of learnable parameters in the MLP model $f_\theta$, $\mathbf{S}_n$ is the adaptive sampling strategy after transformation (i.e., $\mathbf{S}_n = g_\lambda(\mathbf{S})$), and $\alpha$ is a hyper-parameter that controls the relative importance between Wasserstein distance/node-level ranking loss and GW distance/edge-level ranking loss. Through alternating optimization, both OT mapping $\mathbf{S}$ and node embeddings $\mathbf{E}_1, \mathbf{E}_2$ can be optimized in a mutually beneficial manner. The overall algorithm is summarized in Algorithm 1 in Appendix A.

## 3.3 Unified Model Training

In this subsection, we present the model training framework under a unified objective function. Through a simple yet effective transformation, the FGW distance and multi-level ranking loss are combined into a single objective (Subsection 3.3.1), which can be efficiently optimized using an alternating optimization scheme with guaranteed convergence (Subsection 3.3.2).

*3.3.1 Unifying FGW Distance and Multi-level Ranking Loss.* The FGW distance is shown to be a powerful objective for network alignment, and has been adopted by several works [3, 5, 25, 31] to

---

$^2$We use $\mathbf{C}_i$ to encode edge information in two graphs with $\mathbf{C}_i(a, b) = 0, \forall (a, b) \notin \mathcal{E}_i$.

supervise embedding learning. In general, based on the Envelop theorem [1], existing methods based on the FGW objective [3, 5, 25, 31] optimize the cost matrices under the fixed OT mapping $\mathbf{S}$, whose gradients further guide the learning of feature encoder $f_\theta$. However, due to the non-negativity of $\mathbf{S}$, directly minimizing FGW distance leads to trivial solutions where cost matrices $\mathbf{M}, \mathbf{C}_1, \mathbf{C}_2$ become zero matrices, hence leading to embedding collapse as illustrated in the following proposition.

PROPOSITION 1. (EMBEDDING COLLAPSE). *Given two networks $\mathcal{G}_1, \mathcal{G}_2$, directly optimizing feature encoder $f_\theta$ with the FGW distance leads to embedding collapse, that is $\mathbf{E}_1(x) = \mathbf{E}_2(y), \forall x \in \mathcal{G}_1, y \in \mathcal{G}_2$, where $\mathbf{E}_1 = f_\theta(\mathcal{G}_1), \mathbf{E}_2 = f_\theta(\mathcal{G}_2)$.*

The proof can be be found in Appendix B. In general, due to the non-negativity of FGW distance [26], the minimal FGW distance of value zero is achieved by simply projecting all nodes to identical embeddings, hence significantly degrading the discriminating power of learned embeddings.

To alleviate embedding collapse, we propose a transformation $g_\lambda : \mathbb{R}^{n_1 \times n_2}_{\geq 0} \to \mathbb{R}^{n_1 \times n_2}$ to transform the non-negative OT mapping $\mathbf{S}$ into an adaptive node sampling matrix $\mathbf{S}_n = g_\lambda(\mathbf{S})$ to discern the positive samples from the negative ones together with sampling weights. In this work, we adopt $g_\lambda(\mathbf{S}) = \mathbf{S} - \lambda \mathbf{1}_{n_1 \times n_2}$, where $\lambda$ is a learnable transformation parameter. The rationale behind such design is to find the optimal threshold $\lambda$ to distinguish between positive and negative pairs automatically. Moreover, as the absolute value of $\mathbf{S}_n(x, y)$ indicates the certainty of sampling node pair $(x, y)$ as positive/negative pairs, it helps distinguish easy and hard samples for the ranking loss. Equipped with such adaptive sampling matrix $\mathbf{S}_n$, we quantitatively attribute the effectiveness of FGW distance from the following two aspects: *node-level ranking* and *edge-level ranking*.

**Wasserstein Distance as Node-Level Ranking Loss.** Equipped with the sampling strategy $\mathbf{S}_n$, the Wasserstein distance term can be reformulated as a node-level ranking loss as follows

$$\mathcal{J}_{\text{w}} = \sum_{(x, y) \in \mathcal{V}_1 \times \mathcal{V}_2} \mathbf{M}(x, y) \mathbf{S}_n(x, y)$$

$$= \sum_{(x, y_p) \in \mathcal{R}^+} \mathbf{M}(x, y_p) |\mathbf{S}_n(x, y_p)| - \sum_{(x, y_n) \in \mathcal{R}^-} \mathbf{M}(x, y_n) |\mathbf{S}_n(x, y_n)| \tag{6}$$

where $\mathcal{R}^+ = \{(x, y_p) | \mathbf{S}_n(x, y_p) \geq 0\}, \mathcal{R}^- = \{(x, y_n) | \mathbf{S}_n(x, y_n) < 0\}$.

$\mathcal{R}^+$ and $\mathcal{R}^-$ are the sets of positive and negative node pairs, respectively. Therefore, Eq. (6) can be viewed as a weighted ranking loss function at the *node* level, where the sign of $\mathbf{S}_n(x, y)$ is used to distinguish between positive and negative node pairs and the sampling weight $|\mathbf{S}_n(x, y)|$ indicates the certainty of the sampled positive/negative node pair. For example, $(x, y)$ is regarded as an uncertain pair and should contribute little to the ranking loss if $\mathbf{S}(x, y)$ is close to the threshold $\lambda$. Similarly, if $\mathbf{S}(x, y)$ is far away from $\lambda$, the relationship between $(x, y)$ is more certain and should contribute more to the ranking loss. Therefore, $\mathbf{S}_n$ directly models *all* cross-network pairs $(x, y) \in \mathcal{V}_1 \times \mathcal{V}_2$ with noise-reduced certainty values. To this point, we provide a unified view of the Wasserstein distance and the node-level ranking loss.

Another limitation of the existing ranking loss is that it only considers node relationships while ignores the modeling of edge

relationships, hence may fall short in preserving graph structure in the node embedding space [25, 32]. To address this issue, we also introduce a novel ranking loss function at edge-level and unify it with the GW distance.

**Gromov-Wasserstein Distance as Edge-Level Ranking Loss.** The GW distance term can be reformulated as an edge-level ranking loss as follows

$$
\begin{aligned}
\mathcal{J}_{\text{gw}} &= \sum_{\substack{x,x' \in \mathcal{G}_1, \\ y,y' \in \mathcal{G}_2}} |C_1(x,x') - C_2(y,y')|^2 S_n(x,y) S_n(x',y') \\
&= \sum_{(e_{x,x'}, e_{y_p,y'_p}) \in \mathcal{T}^+} d_e(e_{x,x'}, e_{y_p,y'_p}) |S_e(e_{x,x'}, e_{y_p,y'_p})| - \\
&\quad \sum_{(e_{x,x'}, e_{y_n,y'_n}) \in \mathcal{T}^-} d_e(e_{x,x'}, e_{y_n,y'_n}) |S_e(e_{x,x'}, e_{y_n,y'_n})|
\end{aligned} \quad (7)
$$

$$
\text{where} \begin{cases}
d_e(e_{x,x'}, e_{y,y'}) = |C_1(x,x') - C_2(y,y')|^2 \\
S_e(e_{x,x'}, e_{y,y'}) = S_n(x,y) S_n(x',y') \\
\mathcal{T}^+ = \{(e_{x,x'}, e_{y_p,y'_p}) | S_e(e_{x,x'}, e_{y_p,y'_p}) \geq 0\} \\
\mathcal{T}^- = \{(e_{x,x'}, e_{y_n,y'_n}) | S_e(e_{x,x'}, e_{y_n,y'_n}) < 0\}
\end{cases},
$$

where $e_{x,x'}$ is the edge between $x$ and $x'$, $d_e$ measures the distance between two edges, and $\mathcal{T}^+, \mathcal{T}^-$ are the sets of positive and negative edge pairs sampled by the edge sampling strategy $S_e$. Similar to Eq. (6), Eq. (7) is a weighted ranking loss at the *edge* level, where the sign of $S_e(e_{x,x'}, e_{y,y'})$ distinguishes between positive and negative edge pairs and the sampling weight $|S_e(e_{x,x'}, e_{y,y'})|$ indicates the certainty of the sampled positive/negative edge pair. In fact, from the view of line graph [32], where edges in the original graph are mapped into nodes in the line graph and vice versa, the edge ranking loss in the original graph can be interpreted as the node ranking loss in the corresponding line graph. Equipped with ranking loss functions at both node and edge level, Eq. (5) is capable of preserving multi-level graph structure in the node embedding space.

*3.3.2 Optimization.* Combining the node-level ranking loss (Wasserstein distance) and edge-level ranking loss (GW distance) gives the unified optimization objective of JOENA for both embedding learning and OT optimization as Eq. (5). To optimize this objective, we adopt an alternating optimization scheme where the parameters of feature encoder $f_\theta$, transformation parameter $\lambda$, and OT mapping $S$ are optimized iteratively.

Specifically, for the $k$-th iteration in alternating optimization, we first fix the feature encoder $f_\theta^{(k)}$ and the transformation parameter $\lambda^{(k)}$, and optimize Eq. (5) w.r.t $S$ by the proximal point method [30]. Due to the non-convexity of the objective, proximal point method decomposes the non-convex problem into a series of convex subproblems plus an additional Bregman divergence between two consecutive solutions, where each subproblem can be formulated as follows

$$
S^{(t+1)} = \underset{S \in \Pi(\mu_1, \mu_2)}{\arg\min} \mathcal{J}(S; \lambda^{(k)}, \theta^{(k)}) + \gamma_p \text{Div}(S \| S^{(t)}), \quad (8)
$$

where $t$ is the number of proximal point iteration, $\gamma_p$ is the weight for the proximal operator, and Div is the Bregman divergence between two OT mappings. Then, the resulting subproblem in Eq. (8)

can be transformed into an entropy-regularized OT problem as

$$
\min_{S \in \Pi(\mu_1, \mu_2)} \underbrace{\langle C_{\text{total}}^{(t)}, S \rangle + \gamma_p \langle \log S, S \rangle}_{\text{entropy-regularized OT}} - \underbrace{\left\langle (1-\alpha)M + \alpha L_{\text{gw}}^{(t)}, \lambda \right\rangle}_{\text{constant}}
$$

$$
\text{where} \begin{cases}
C_{\text{total}}^{(t)} = (1-\alpha)M + \alpha L_{\text{gw}}^{(t)} - \gamma_p \log S^{(t)} \\
L_{\text{gw}}^{(t)} = C_1^2 S_n^{(t)} \mathbf{1}_{n_2 \times n_2} + \mathbf{1}_{n_1 \times n_1} S_n^{(t)} C_2^{2\top} - 2C_1 S_n^{(t)} C_2^\top. \\
S_n^{(t)} = S^{(t)} - \lambda^{(k)} \mathbf{1}_{n_1 \times n_2}
\end{cases} \quad (9)
$$

Note that $S^{(t)}$ is the OT mapping from last proximal point iteration and remain fixed in the above equation. Therefore, the objective function of each proximal point iteration in Eq. (9) is essentially an entropy-regularized OT problem with a fixed transport cost $C_{\text{total}}^{(t)}$ minus a constant term that does not affect the optimization. Following the Sinkhorn algorithm [18], Eq. (9) can be solved efficiently.

Then, we fix the feature encode $f_\theta^{(k)}$ and OT mapping $S^{(k+1)}$, and optimize Eq. (5) w.r.t the transformation parameter $\lambda$. Since the objective function is quadratic w.r.t. $\lambda$, the closed-form solution for $\lambda^{(k+1)}$ can be obtained by setting $\partial \mathcal{J}/\partial \lambda = 0$ as follows

$$
\lambda^{(k+1)} = \frac{(1-\alpha)\mathcal{K}_1 + \alpha \mathcal{K}_2}{2\alpha \mathcal{K}_3}
$$

$$
\text{where} \begin{cases}
\mathcal{K}_1 = \sum_{x \in \mathcal{G}_1, y \in \mathcal{G}_2} M(x, y; \theta^{(k)}) \\
\mathcal{K}_2 = \sum_{\substack{x,x' \in \mathcal{G}_1 \\ y,y' \in \mathcal{G}_2}} d_e\left(e_{x,x'}, e_{y,y'}; \theta^{(k)}\right)\left(S^{(k+1)}(x, y) + S^{(k+1)}(x', y')\right) \\
\mathcal{K}_3 = \sum_{\substack{x,x' \in \mathcal{G}_1 \\ y,y' \in \mathcal{G}_2}} d_e\left(e_{x,x'}, e_{y,y'}; \theta^{(k)}\right)
\end{cases} \quad (10)
$$

Finally, to optimize the feature encoder $f_\theta$, we fix the transformation parameter $\lambda^{(k+1)}$ and the OT mapping $S^{(k+1)}$ to optimize Eq. (5) w.r.t $\theta$ via stochastic gradient descent (SGD), that is

$$
\theta^{(k+1)} = \arg\min_\theta \mathcal{J}(\theta; S^{(k+1)}, \lambda^{(k+1)}). \quad (11)
$$

As we will show later, by iteratively applying Eq. (8)-(11), the objective function in Eq. (5) converges under the alternating optimization scheme. Besides, it is worth noting that alternating optimization is only used for model training, while model inference only requires one-pass, i.e., the forward pass of MLP and the proximal point method for OT optimization, allowing JOENA to scale efficiently to large networks.

## 3.4 Proof and Analysis

In this subsection, we provide theoretical analysis of the proposed JOENA. Without loss of generality, we assume that graphs share comparable numbers of nodes (i.e., $O(n_1) \approx O(n_2) \approx O(n)$) and edges (i.e., $O(m_1) \approx O(m_2) \approx O(m)$). We first provide the convergence analysis of JOENA, followed by complexity analysis.

THEOREM 1. (CONVERGENCE OF JOENA) *The unified objective for JOENA in Eq. (5) is non-increasing and converges along the alternating optimization.*

PROPOSITION 2. (COMPLEXITY OF JOENA) *The overall time complexity of JOENA is $O\left(KTmn + KTNn^2\right)$ at the training phase and $O\left(Tmn + TNn^2\right)$ at the inference phase, where $K, T, N$ denote the*

*number of iterations for alternating optimization, proximal point iteration, and Sinkhorn algorithm, respectively.*

All the proofs can be found in Appendix B. In general, the alternating optimization scheme generates a series of non-increasing objective functions with a bounded minimum hence achieving guaranteed convergence. In addition, as we can see, JOENA achieves fast inference with linear complexity w.r.t the number of edges and quadratic complexity w.r.t the number of nodes.

## 4 Experiments

In this section, we carry out extensive experiments and analyses to evaluate the proposed JOENA from the following aspects:

- **Q1.** How effective is the proposed JOENA?
- **Q2.** How efficient and scalable is the proposed JOENA?
- **Q3.** How robust is JOENA against graph noises?
- **Q4.** How do OT and embedding learning benefit each other?
- **Q5.** To what extent does the OT-based sampling strategy surpass the hand-crafted strategies?

### 4.1 Experiment Setup

**Datasets.** Our method is evaluated on both plain and attributed networks summarized in Table 5. We use 20% ground-truth as the anchor nodes and the rest 80% of the ground-truth for testing. Detailed descriptions and experimental settings are included in Appendix C. We will release our code upon publication.

**Baselines.** JOENA is compared with the following three groups of methods, including (1) Consistency-based methods: IsoRank [22] and FINAL [40], (2) Embedding-based methods: REGAL [11], DANA [12], NetTrans [44], BRIGHT [34], NeXtAlign [41], and WL-Align [13], and (3) OT-based methods: WAlign [10], GOAT [21], PARROT [38], and SLOTAlign [25]. To ensure a fair and consistent comparison, for all unsupervised baselines, we introduce the supervision information in the same way as JOENA by concatenating the RWR scores w.r.t the anchor nodes with the node input features.

**Metrics.** We adopt two commonly used metrics Hits@K and Mean Reciprocal Rank (MRR) to evaluate model performance. Specifically, given $(x, y) \in \mathcal{S}_{\text{test}}$ where $\mathcal{S}_{\text{test}}$ denotes the set of testing node pairs, if node $y \in \mathcal{G}_2$ is among the top-$K$ most similar nodes to node $u \in \mathcal{G}_1$, we consider it as a hit. Then, Hits@K is computed by Hits@$K = \frac{\text{\# of hits}}{|\mathcal{S}_{\text{test}}|}$. MRR is computed by the average of the reciprocal alignment ranking of all testing node pair, i.e., MRR = $\frac{1}{|\mathcal{S}_{\text{test}}|} \sum_{(x,y) \in \mathcal{S}_{\text{test}}} \frac{1}{\text{rank}(x,y)}$.

### 4.2 Effectiveness Results

We evaluate the alignment performance of JOENA, and the results on plain and attributed networks are summarized in Table 2 and 3, respectively. Compared with consistency and embedding-based methods, JOENA achieves up to 31% and 22% improvement in MRR over the best-performing baseline on plain and attributed network tasks, respectively, which indicates that JOENA is capable of learning noise-reduced node mapping beyond local graph geometry and consistency principles thanks to the OT component. Compared with OT-based methods, JOENA achieves a significant outperformance compared with the best competitor PARROT [38], achieving up to 16% and 6% improvement in MRR on plain and attributed

**Table 2: Performance on plain network alignment.**

| Dataset | Foursquare-Twitter | | | ACM-DBLP | | | Phone-Email | | |
|---|---|---|---|---|---|---|---|---|---|
| Metrics | Hits@1 | Hits@10 | MRR | Hits@1 | Hits@10 | MRR | Hits@1 | Hits@10 | MRR |
| IsoRank | 0.028 | 0.189 | 0.087 | 0.157 | 0.629 | 0.297 | 0.023 | 0.133 | 0.060 |
| FINAL | 0.040 | 0.236 | 0.100 | 0.196 | 0.692 | 0.354 | 0.031 | 0.215 | 0.099 |
| DANA | 0.042 | 0.160 | 0.082 | 0.343 | 0.559 | 0.316 | 0.033 | 0.206 | 0.095 |
| NetTrans | 0.086 | 0.270 | 0.145 | 0.410 | 0.801 | 0.540 | 0.065 | 0.119 | 0.155 |
| BRIGHT | 0.091 | 0.268 | 0.149 | 0.394 | 0.809 | 0.534 | 0.043 | 0.255 | 0.113 |
| NeXtAlign | 0.101 | 0.279 | 0.158 | 0.459 | 0.861 | 0.594 | 0.063 | 0.424 | 0.195 |
| WL-Align | 0.253 | 0.343 | 0.285 | 0.542 | 0.781 | 0.629 | 0.121 | 0.409 | 0.214 |
| WAlign | 0.077 | 0.258 | 0.135 | 0.342 | 0.794 | 0.481 | 0.046 | 0.308 | 0.131 |
| PARROT | 0.245 | 0.409 | 0.304 | 0.619 | 0.912 | 0.719 | 0.323 | 0.749 | 0.469 |
| JOENA | **0.403** | **0.576** | **0.464** | **0.635** | **0.933** | **0.736** | **0.384** | **0.809** | **0.527** |

**Table 3: Performance on attributed network alignment.**

| Dataset | Cora1-Cora2 | | | ACM(A)-DBLP(A) | | | Douban | | |
|---|---|---|---|---|---|---|---|---|---|
| Metrics | Hits@1 | Hits@10 | MRR | Hits@1 | Hits@10 | MRR | Hits@1 | Hits@10 | MRR |
| FINAL | 0.710 | 0.881 | 0.773 | 0.398 | 0.833 | 0.542 | 0.468 | 0.914 | 0.625 |
| REGAL | 0.511 | 0.591 | 0.542 | 0.511 | 0.591 | 0.542 | 0.099 | 0.274 | 0.153 |
| NetTrans | 0.989 | 0.999 | 0.993 | 0.692 | 0.938 | 0.779 | 0.210 | 0.213 | 0.332 |
| BRIGHT | 0.839 | 0.992 | 0.905 | 0.470 | 0.857 | 0.603 | 0.282 | 0.641 | 0.397 |
| NeXtAlign | 0.439 | 0.703 | 0.538 | 0.486 | 0.867 | 0.615 | 0.245 | 0.655 | 0.385 |
| WAlign | 0.824 | 0.997 | 0.901 | 0.377 | 0.779 | 0.501 | 0.236 | 0.533 | 0.341 |
| PARROT | 0.996 | **1.000** | 0.998 | 0.721 | 0.960 | 0.806 | 0.696 | 0.981 | 0.789 |
| SLOTAlign | 0.985 | 0.997 | 0.990 | 0.663 | 0.879 | 0.740 | 0.486 | 0.762 | 0.582 |
| JOENA | **0.999** | **1.000** | **0.999** | **0.767** | **0.967** | **0.839** | **0.761** | **0.986** | **0.851** |

networks. Such outperformance demonstrates the effectiveness of the learnable transport costs encoded by learnable node embeddings. Moreover, the performance improvement over WAlign [10] and SLOTAlign [25] indicates that JOENA successfully avoids embedding collapse thanks to the learnable transformation $g_\lambda$ on OT mapping and the resulting adaptive sampling strategy $S_n$.

### 4.3 Robustness Results

To show the robustness of the proposed JOENA , we evaluate the performance of JOENA under two kinds of graph noises: *structural* noise and *attribute* noise.

*4.3.1 Robustness against Structural Noises.* We first evaluate the robustness of JOENA against structural (edge) noises. Specifically, for edge noise level $p$, we randomly flip $p$% entries in the adjacency matrix, i.e., randomly add/delete edges [25]. Evaluations are conducted on the plain network Phone-Email to eliminate the potential interference from node attributes. The results are shown in Figure 3a.

Compared to other baselines, the performance of JOENA consistently achieves the highest MRR in all cases. More importantly, thanks to the direct modeling and noise-reduced property of OT, we observe a much slower degradation of the MRR when the noise level increases, validating the robustness of JOENA against graph structural noises. Furthermore, embedding-based methods without OT (i.e., WLAlign [13], NeXtAlign [41], BRIGHT [34]) degrades much faster than methods with OT (i.e., JOENA, PARROT [38]), demonstrating that embedding-based methods are more sensitive to structural noise due to indirect modeling.

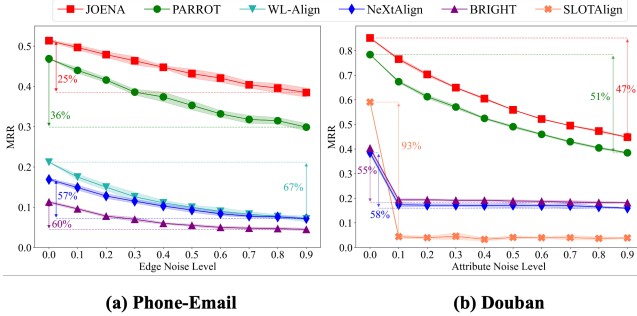

**(a) Phone-Email**        **(b) Douban**

**Figure 3: Performance comparison of five alignment methods under different levels of structure and attribute noise.**

*4.3.2 Robustness against Attribute Noises.* We also evaluate the robustness of JOENA against attribute noises. Specifically, for attribute level $p$, we randomly flip $p\%$ entries in the attribute matrix [28]. The results are shown in Figure 3b.

Compared to baselines, the performance of JOENA consistently achieves the best performance, as well as the mildest degradation when attribute noise level increases, demonstrating the robustness of JOENA against node attribute noises. Besides, the performance of embedding-based methods degrades more severely than JOENA which further illuminates the deficiency of indirect modeling.

## 4.4 Scalability Results

We compare the scalability of the propose JOENA with that of OT-based methods, including GOAT [21], PARROT [38], and SLOTAlign [25]. We record the inference time as the number of edges increases, and the results are shown in Figure 4. For networks with 20,000 edges, JOENA runs 20 times faster than SLOTAlign. Under 300-second running time limit, JOENA can process networks 5 times the size of SLOTAlign. Besides, we observe that JOENA runs slightly faster than the pure OT-based method PARROT. For one thing, we attribute such slight improvement to the lightweight MLP for embedding learning, as PARROT requires hand-crafted embeddings that may be computationally-heavy. For another, better cost design based on learnable embeddings may also benefit the converegence of OT optimization, hence achieving faster computation.

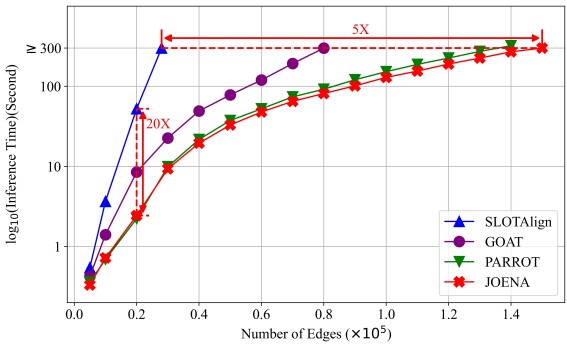

**Figure 4: Scalability results. JOENA achieves the best scalability results with up to 20× speed-up in inference time and up to 5× scale-up in network size.**

**Table 4: Mutual benefits of embedding and OT learning**

| Dataset | Foursquare-Twitter | | | ACM-DBLP | | | Phone-Email | | |
|---|---|---|---|---|---|---|---|---|---|
| Metrics | Hits@1 | Hits@10 | MRR | Hits@1 | Hits@10 | MRR | Hits@1 | Hits@10 | MRR |
| Emb | 0.079 | 0.244 | 0.134 | 0.401 | 0.798 | 0.534 | 0.063 | 0.358 | 0.164 |
| Emb(OT) | **0.090** | **0.255** | **0.140** | **0.406** | **0.807** | **0.538** | **0.078** | **0.373** | **0.173** |
| OT | 0.243 | 0.407 | 0.298 | 0.600 | 0.916 | 0.707 | 0.224 | 0.581 | 0.343 |
| JOENA | **0.403** | **0.576** | **0.464** | **0.635** | **0.933** | **0.736** | **0.384** | **0.809** | **0.527** |
| OT ⊙ Emb | 0.243 | 0.407 | 0.297 | 0.601 | 0.916 | 0.707 | 0.224 | 0.593 | 0.337 |
| OT + Emb | 0.244 | 0.408 | 0.299 | 0.600 | 0.917 | 0.707 | 0.226 | 0.583 | 0.345 |

## 4.5 Further Analysis

*4.5.1 Mutual Benefits of OT and Embedding Learning.* To verify the mutual benefits of OT and embedding learning, we compare the performance of JOENA against different variants on three real-world networks. Specifically, we consider the following variants: (1) Emb infers node alignments by node embeddings learned under the sampling strategy from BRIGHT [34]; (2) Emb(OT) infers node alignments by node embeddings learned under our OT-based sampling strategy; (3) OT infers node alignments by the OT mapping with cost matrices based on RWR encoding; (4) JOENA infers node alignments by OT mapping with learnable cost matrices; (5) OT⊙Emb infers node alignments by the Hadamard product of OT mapping and the inner product of node embeddings; (6) OT+Emb infers node alignments by the sum of OT mapping and the inner product of node embeddings.

The results are shown in Table 4. Firstly, we observe a consistent outperformance of Emb(OT) compared to Emb, showing that the proposed OT-based sampling strategy improves the quality of node embeddings compared to existing sampling strategies. Besides, comparing OT to JOENA, without learnable cost matrices, OT drops up to 16% in Hits@1 compared to JOENA, indicating that the cost design on learnable node embeddings improves the performance of OT optimization by a significant margin. Furthermore, we compare the performance of JOENA to OT⊙Emb and OT+Emb, both of which naively integrate the OT and embedding alignments learned separately. It is shown that both OT⊙Emb and OT+Emb achieves similar performance as OT and outperforms Emb. For one thing, this suggests that the outperformance of JOENA largely attributes to the OT alignment, which provides a more denoised alignment compared with embedding alignment. For another, naively combining the alignment matrices of embedding or OT-based method at the final stage hardly improves the alignment quality, and it is necessary to combine both components during training.

*4.5.2 OT-based Sampling Strategy.* We also carry out studies on the effectiveness of the OT-based sampling strategy $g_\lambda(\mathbf{S})$. As shown in Figure 6, we report the MRR under different $\lambda$ with the learned $\lambda$ annotated. It is shown that JOENA achieves the best performance under the learned $\lambda$. Besides, we observe a significant performance drop when $\lambda$ is not properly selected. This is due to the fact that when $\lambda$ is too small/large, most pairs will be sampled as positive/negative pairs exclusively, which further destroy the embedding space (i.e., embedding collapse). To validate this point, we visualize how the embedding space changes along optimization. As shown in Figure 5a, when setting $\lambda = 0$, MRR gradually decreases and the learned embeddings collapse into an identical point along

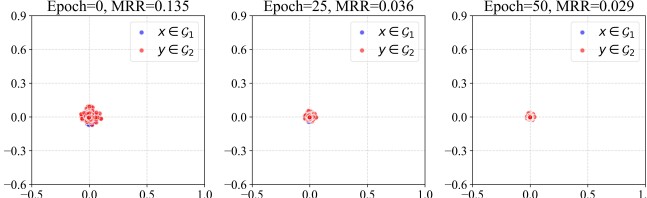

(a) Node embeddings learned via original OT objective, i.e., $\lambda = 0$.

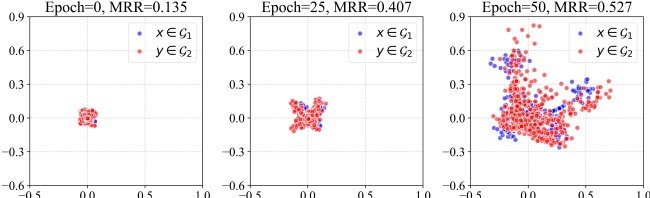

(b) Node embeddings learned via OT objective with optimal $\lambda$.

Figure 5: Evolution of node embeddings along training: (a) directly applying FGW distance for embedding learning leads to embedding collapse and MRR degradation; (b) utilizing FGW distance with transformed $S_n$ leads to discriminating embeddings and MRR improvement.

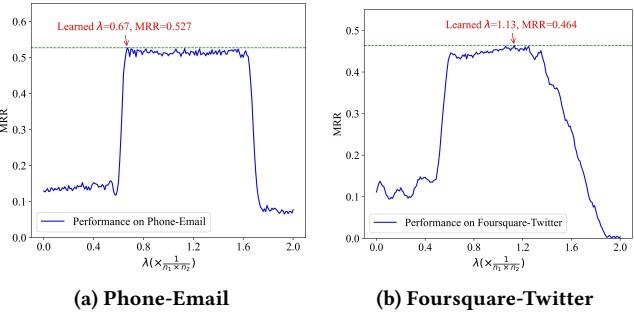

(a) Phone-Email      (b) Foursquare-Twitter

Figure 6: MRR with different $\lambda$. Our learned $\lambda$ consistently achieves the best MRR on both datasets.

optimization. On the contrary, as shown in Figure 5b, JOENA is able to learn the optimal $\lambda$, under which, MRR gradually increases and node embeddings are well separated in the embedding space.

## 5 Related Works

### 5.1 Network Alignment

Traditional network alignment methods are often built upon alignment consistency principle, which assumes that the topology and/or attributes of neighboring nodes to be consistent across networks [22, 40, 42]. IsoRank [22] conducts random walk on the product graph to achieve consistency in graph topology. FINAL [40] interprets IsoRank as an optimization problem and further introduces consistency at attribute level to handle alignment on attributed networks. MOANA [42] aligns networks at multiple granularities to achieve better scalability. However, the consistency assumption only considers node relationships within a local neighborhood, ignoring the overall graph geometry from a global perspective [38].

Another line of works aims to learn informative node embeddings in a unified space to infer alignment. REGAL [11] conducts matrix factorization on cross-network similarity matrix for node embedding learning. DANA [12] learns domain-invariant embeddings for network alignment via adversarial learning. NetTrans [44] aligns networks based on nonlinear network transformation. BRIGHT [34] bridges the consistency and embedding-based alignment methods, and NeXtAlign [41] further balances between the alignment consistency and disparity by crafting the sampling strategies. CPUGA [16] designs a non-sampling model to progressively select potential positive node pairs. WL-Align [13] utilizes cross-network Weisfeiler-Lehman relabeling to learn proximity-preserving embeddings. More related works on network alignment are reviewed in [9].

### 5.2 Optimal Transport on Graphs

OT has recently gained increasing attention in graph and Web mining. The key idea is to represent graphs as distributions over the node sets and minimize the total transportation distance based on cost functions defined over the two distributions. However, the effectiveness of most OT-based alignment methods depends largely on the pre-defined cost function restricted to specific graphs. For example, [10, 15, 17] represent graphs as distributions of filtered graph signals, focusing on one specific graph property, while other cost designs are mostly based on node attributes [3] or graph structures [21]. PARROT [38] integrates various graph properties and consistency principles via a linear combination, which however requires arduous parameter tuning for different graphs.

More recent works have been seeking to combine embedding and OT-based alignment methods to supervise embedding learning for better cost design. GOT [3] adopts a neural network model to encode transport cost at both node and edge levels. GWL [31] learns graph matching and node embeddings jointly in a Gromov-Wasserstein learning framework. SLOTAlign [25] utilizes a parameter-free GNN model to encode the GW distance between two graph distributions. CombAlign [5] further proposes to combine the embeddings and OT-based alignment via an ensemble framework.

## 6 Conclusions

In this paper, we study the semi-supervised network alignment problem by combining embedding and OT-based alignment methods in a mutually beneficial manner. To improve embedding learning via OT, we propose a learnable transformation on OT mapping to obtain an adaptive sampling strategy directly modeling all cross-network node relationships. To improve OT optimization via embedding, we utilize the learned node embeddings to achieve more expressive OT cost design. We further show that the FGW distance can be neatly unified with a multi-level ranking loss at both node and edge levels. Based on these, a unified framework named JOENA is proposed to learn node embeddings and OT mappings in a mutually beneficial manner. Extensive experiments show that JOENA consistently outperforms the state-of-the-art in both effectiveness and scalability by a significant margin, achieving up to 16% performance improvement and up to 20× speedup.

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

## A  Algorithm

We present the overall algorithm of JOENA as Algorithm 1.

---

**Algorithm 1** Joint OT and embedding learning (JOENA)

---

**Input:** (1) networks $\mathcal{G}_1 = (\mathbf{A}_1, \mathbf{X}_1)$, $\mathcal{G}_2 = (\mathbf{A}_2, \mathbf{X}_2)$, (2) anchor node set $\mathcal{L}$, (3) parameters $\alpha, \beta, \gamma_p$
**Output:** the alignment matrix $\mathbf{S}$.
1: Initialize $\boldsymbol{\mu}_1 = \frac{\mathbf{1}_{n_1}}{n_1}, \boldsymbol{\mu}_2 = \frac{\mathbf{1}_{n_2}}{n_2}, \mathbf{S}^{(1)} = \boldsymbol{\mu}_1 \boldsymbol{\mu}_2^{\top}, \lambda^{(1)} = \frac{1}{n_1 \times n_2}$;
2: Compute RWR embedding matrices $\mathbf{R}_1, \mathbf{R}_2$ by Eq. (3);
3: Concatenate node attributes $\mathbf{X}_1 = [\mathbf{R}_1 || \mathbf{X}_1]$, $\mathbf{X}_2 = [\mathbf{R}_2 || \mathbf{X}_2]$
4: **for** $k = 1, ..., K$ **do**
5:   Update node embeddings $\mathbf{E}_1^{(k)} = f_{\theta}^{(k)}(\mathbf{X}_1)$, $\mathbf{E}_2^{(k)} = f_{\theta}^{(k)}(\mathbf{X}_2)$;
6:   Update cost matrices $\mathbf{M}^{(k)}, \mathbf{C}_1^{(k)}, \mathbf{C}_2^{(k)}$ by Eq. (4);
7:   Update OT mapping $\mathbf{S}^{(k+1)}$ by proximal point method in Eq. (8);
8:   Update transformation parameter $\lambda^{(k+1)}$ by Eq. (10);
9:   Update $\theta^{(k+1)}$ by SGD in Eq. (11);
10: **end for**
11: **return** alignment matrix $\mathbf{S}^{(K+1)}$.

---

## B  Proof

### B.1  Proof of Proposition 1

PROPOSITION. (EMBEDDING COLLAPSE). *Given two networks* $\mathcal{G}_1, \mathcal{G}_2$, *directly optimizing feature encoder* $f_{\theta}$ *with the FGW distance leads embedding collapse, that is* $\mathbf{E}_1(x) = \mathbf{E}_2(y), \forall x \in \mathcal{G}_1, y \in \mathcal{G}_2$, *where* $\mathbf{E}_1 = f_{\theta}(\mathcal{G}_1), \mathbf{E}_2 = f_{\theta}(\mathcal{G}_2)$.

PROOF. Firstly, the Wasserstein term can be written as

$$\sum_{x \in \mathcal{G}_1, y \in \mathcal{G}_2} \mathbf{M}^q(x, y) \mathbf{S}(x, y) \tag{12}$$

Due to the non-negativity of $\mathbf{M}$ and $\mathbf{S}$, i.e., $\mathbf{S}(x, y) \geq 0, \mathbf{M}(x, y) \geq 0, \forall x \in \mathcal{G}_1, y \in \mathcal{G}_2$, the Wasserstein term in Eq. (12) has a theoretical minimum of 0. Since Eq. (12) is a linear programming problem w.r.t $\mathbf{S}$ which is computationally demanding to solve, existing works turn to solve the entropy-regularized OT problem to approximate Eq. (12), where the solved $\mathbf{S}$ is strictly positive, i.e. $\mathbf{S}(x, y) > 0, \forall x \in \mathcal{G}_1, y \in \mathcal{G}_2$. We can simply prove by contradiction that Eq. (12) reaches 0 if and only if $\forall x \in \mathcal{G}_1, y \in \mathcal{G}_2, \mathbf{M}(x, y) = 0$. According to the universal approximation theorem [7], such cross-network cost matrix is achievable with a MLP. Therefore, optimizing Eq. (12) under a node mapping matrix $\mathbf{S}$ will lead to collapsed node embeddings across two networks, i.e., $\mathbf{E}_1(x) = \mathbf{E}_2(y), \forall x \in \mathcal{G}_1, y \in \mathcal{G}_2$.

Secondly, the GW term can be formulated as

$$\sum_{\substack{x_1, x_2 \in \mathcal{G}_1 \\ y_1, y_2 \in \mathcal{G}_2}} |\mathbf{C}_1(x_1, x_2) - \mathbf{C}_2(y_1, y_2)|^q \mathbf{S}(x_1, y_1) \mathbf{S}(x_2, y_2). \tag{13}$$

Similarly, due to the non-negativity of $|\mathbf{C}_1(x_1, x_2) - \mathbf{C}_2(y_1, y_2)|^q$ and the positivity of $\mathbf{S}(x_1, y_1)\mathbf{S}(x_2, y_2)$, the GW term in Eq (13) has a theoretical minimum of 0 if and only if $\forall x_1, x_2 \in \mathcal{G}_1, y_1, y_2 \in \mathcal{G}_2, |\mathbf{C}_1(x_1, x_2) - \mathbf{C}_2(y_1, y_2)|^q = 0$. Since $\mathbf{C}_1(x, x) = \mathbf{C}_2(y, y) = 0, \forall x_1, x_2 \in \mathcal{G}_1, y_1, y_2 \in \mathcal{G}_2, \mathbf{C}_1(x_1, x_2) = \mathbf{C}_2(y_1, y_2) = 0$, which essentially means the embeddings of all nodes in $\mathcal{G}_1$ ($\mathcal{G}_2$) collapse into a single point, i.e., $\mathbf{E}_1(x_1) = \mathbf{E}_1(x_2), \mathbf{E}_2(y_1) = \mathbf{E}_2(y_2), \forall x_1, x_2 \in$

$\mathcal{G}_1, y_1, y_2 \in \mathcal{G}_2$. By combining Eq. (12) and Eq. (13), the Wasserstein term further causes the embedding of all nodes in both networks to collapse into a single point, i.e., $\mathbf{E}_1(x) = \mathbf{E}_2(y), \forall x \in \mathcal{G}_1, y \in \mathcal{G}_2$. Therefore, directly optimizing feature encoder with the FGW distance leads embedding collapse. □

### B.2  Proof of Theorem 1

THEOREM. (CONVERGENCE OF JOENA) *The unified objective for JOENA in Eq. (5) is non-increasing and converges along the alternating optimization.*

PROOF. We first prove Eq. (5) is bounded by a minimum value. We make a common assumption that the parameter set $\theta$ of the MLP is bounded [25]. Since $\mathbf{S} \in \Pi(\boldsymbol{\mu}_1, \boldsymbol{\mu}_2)$ is bounded as well, we only need to prove that Eq. (5) is bounded w.r.t $\lambda$, which is essentially a quadratic function with a non-negative coefficient for the quadratic term, i.e.,

$$\sum_{\substack{x_1, x_2 \in \mathcal{G}_1 \\ y_1, y_2 \in \mathcal{G}_2}} |\mathbf{C}_1(x_1, x_2) - \mathbf{C}_2(y_1, y_2)|^2 \geq 0$$

By solving $\lambda$ based on $\partial \mathcal{J}/\partial \lambda = 0$ according to Eq. (10), we have the optimal $\lambda^*$ minimizing Eq. (5) as follows

$$\min_{\lambda} \mathcal{J}(\mathbf{S}, \lambda, \theta) = \mathcal{J}(\mathbf{S}, \lambda^*, \theta).$$

Since both $\theta$ and $\mathbf{S}$ are bounded, there exists a real number $\epsilon \in \mathbb{R}$ satisfying

$$\mathcal{J}(\mathbf{S}, \lambda, \theta) \geq \mathcal{J}(\mathbf{S}, \lambda^*, \theta) > \epsilon$$

In this way, we have prove that Eq. (5) is bounded by a minimum value $\epsilon$.

Then, we prove that Eq. (5) is non-increasing and converges along the alternating optimization, i.e.,

$$\mathcal{J}(\mathbf{S}^{(k+1)}, \lambda^{(k+1)}, \theta^{(k+1)}) \leq \mathcal{J}(\mathbf{S}^{(k)}, \lambda^{(k)}, \theta^{(k)}) \tag{14}$$

To prove Eq. (14), we first show that the OT optimization by proximal point method is non-increasing. Specifically, as proved theoretically in [31], the proximal point method solves Eq. (5) w.r.t $\mathbf{S}$ by decomposing the non-convex objective function into a series of convex approximations, which be viewed as a successive upper-bound minimization [20] problem whose descend property is guaranteed. In this way, we have demonstrated that

$$\mathcal{J}(\mathbf{S}^{(k+1)}, \lambda^{(k)}, \theta^{(k)}) \leq \mathcal{J}(\mathbf{S}^{(k)}, \lambda^{(k)}, \theta^{(k)}) \tag{15}$$

Then, we solve $\lambda^{(k+1)}$ optimally based on the closed-form solution in Eq. (10) with guaranteed global minimum. Therefore, we have

$$\mathcal{J}(\mathbf{S}^{(k+1)}, \lambda^{(k+1)}, \theta^{(k)}) \leq \mathcal{J}(\mathbf{S}^{(k+1)}, \lambda^{(k)}, \theta^{(k)}) \tag{16}$$

Finally, with an appropriate learning rate, the objective of the embedding learning process via SGD is non-increasing at each step, i.e.,

$$\mathcal{J}(\mathbf{S}^{(k+1)}, \lambda^{(k+1)}, \theta^{(k+1)}) \leq \mathcal{J}(\mathbf{S}^{(k+1)}, \lambda^{(k+1)}, \theta^{(k)}) \tag{17}$$

Combining Eq. (15)-(17) gives Eq. (14). In this way, we have proven Theorem 1. □

## B.3 Proof of Proposition 2

PROPOSITION. (COMPLEXITY OF JOENA) *The overall time complexity of JOENA is* $O\left(KTmn + KTNn^2\right)$ *at the training phase and* $O\left(Tmn + TNn^2\right)$ *at the inference phase, where* $K, T, N$ *denote the number of iterations for alternating optimization, proximal point iteration, and Sinkhorn algorithm, respectively.*

PROOF. The time complexity of JOENA includes four components: RWR encoding, MLP computation, calculation of the optimal $\lambda$, and OT optimization. Since $\mathbf{C}_i$ and $\mathbf{W}_i$ are sparse matrices with $O(m)$ non-zero entries, the time complexity of RWR in Eq. (3) is $O(mn)$ [38].

For each iteration of the alternating optimization, the time complexity for forward (backward) propagation of the MLP model $\mathcal{G}_1(\mathcal{G}_2)$ are $O(n|\mathcal{L}|d_1)$ (first layer) and $O(n|\mathcal{L}|d_2)$ (second layer), respectively. For the calculation of the optimal $\lambda$, the time complexity is $O(mn)$ [38]. For the OT optimization, the time complexity is $O(Tmn + TNn^2)$ with $T$ iterations of proximal point method and $N$ Sinkhorn iterations [38].

Combining the above three components gives a total time complexity of $O\left(K(2n|\mathcal{L}|d_1 + 2n|\mathcal{L}|d_2 + (T+1)mn + TNn^2)\right)$ where $K$ is the number of iteration for the alternating optimization. Since $n \gg |\mathcal{L}|, T \gg 1$, the overall training time complexity of JOENA is $O(KTmn + KTNn^2)$. Note that model inference is only one-pass without the alternating optimization, hence the inference time complexity is $O\left(Tmn + TNn^2\right)$. □

## C Experiment Pipeline

**Table 5: Dataset Statistics.**

| Scenarios | Networks | # nodes | # edges | # attributes |
|---|---|---|---|---|
| Plain | Foursquare | 5,313 | 54,233 | 0 |
| | Twitter | 5,120 | 130,575 | 0 |
| | ACM | 9,872 | 39,561 | 0 |
| | DBLP | 9,916 | 44,808 | 0 |
| | Phone | 1,000 | 41,191 | 0 |
| | Email | 1,003 | 4,627 | 0 |
| Attributed | Cora1 | 2,708 | 6,334 | 1,433 |
| | Cora2 | 2,708 | 4,542 | 1,433 |
| | ACM(A) | 9,872 | 39,561 | 17 |
| | DBLP(A) | 9,916 | 44,808 | 17 |
| | Douban(online) | 3,906 | 16,328 | 538 |
| | Douban(offline) | 1,118 | 3,022 | 538 |

**Dataset Descriptions.** The datasets used in our experiments are described as follows.

- **Foursquare-Twitter** [39]: A pair of online social networks with nodes representing users and edges representing follower/followee relationships. Foursquare network contains 5,313 nodes and 5,120 edges. Twitter network contains 5,120 nodes and 130,575 edges. Node attributes are unavailable in both networks. There are 1,609 common users across the two networks that are used as ground-truth.

- **ACM-DBLP** [24]: A pair of undirected co-authorship networks with nodes representing authors and edges representing co-authorship. ACM network contains 9,916 nodes and 44,808 edges. DBLP network contains 9,872 nodes and 39,561 edges. Node attributes are available in both networks, and we use the dataset for both plain and attributed network alignment tasks with the name ACM-DBLP and ACM(A)-DBLP(A), respectively. There are 6,325 common authors across the two networks that are used as ground-truth.

- **Phone-Email** [43]: A pair of communication networks with nodes representing people and edges representing their communications via phone or email. Phone networks contains 1,000 nodes and 41,191 edges. Email networks contains 1,003 nodes and 4,627 edges. Node attributes are unavailable in both networks. There are 1,000 common people across the two networks that are used as ground-truth.

- **Cora1-Cora2** [35]. A citation network with nodes representing publications and edges representing citations among publications. Cora-1 and Cora-2 are two noisy permutation networks generated from the Cora citation network by inserting 10% edges into Cora-1 and deleting 15% edges from Cora-2. Cora-1 contains 2,708 nodes and 6,334 edges. Cora-2 contains 2,708 nodes and 4,542 edges. Both networks contains node attributes which are binary vectors represented by bag-of-words. There are 2,708 common publications across the two networks that are used as ground-truth.

- **Douban** [40]. A pair of social networks with nodes representing users and edges representing user interactions on the website. Online network contains 3,906 nodes and 16,328 edges. Offline network contains 1,118 nodes and 3,022 edges. The node attributes are binary vectors that encodes the location of a user. There are 1,118 common user across the two networks that are used as ground-truth.

Dataset statistics are given in Table 5.

**Machine and Code.** The proposed model is implemented in PyTorch. We use Apple M1 Pro with 16GB RAM to run PARROT, IsoRank, FINAL, and GOAT. We use NVIDIA Tesla V100 SXM2 as GPU for JOENA and other baselines.

**Implementation Details.** Adam optimizer is used with a learning rate of 1e-4 to train the model. The hidden and output dimension is set to 128. The epoch number of JOENA is 50. An overview of other hyperparameters settings for JOENA is shown in Table 6. For all baselines, hyperparameters are set as default in their official code.

**Table 6: Hyperparameters settings**

| Dataset | $\alpha$ | $\beta$ | $\gamma_p$ |
|---|---|---|---|
| Foursquare-Twitter | 0.50 | 0.15 | 1e-3 |
| ACM-DBLP | 0.90 | 0.15 | 5e-3 |
| Phone-Email | 0.75 | 0.15 | 1e-2 |
| ACM(A)-DBLP(A) | 0.90 | 0.15 | 1e-2 |
| Cora1-Cora2 | 0.30 | 0.15 | 5e-4 |
| Douban | 0.75 | 0.15 | 1e-3 |

Received 20 February 2007; revised 12 March 2009; accepted 5 June 2009

