# OpenReview forum: "Joint Optimal Transport and Embedding for Network Alignment"
_ACM.org/TheWebConf/2025/Conference — WWW 2025 Poster_

### Official Review · Reviewer_k34y · 2024-11-23

**Novelty:** 4
**Technical Quality:** 4

**Review:**

**Summary**

This paper proposes JOENA, a unified framework combining embedding-based methods and optimal transport (OT) for network alignment, improving robustness and alignment quality. JOENA enhances performance by using OT for adaptive sampling in embedding learning and embedding-based training for OT cost, achieving significant improvements in alignment accuracy and computational efficiency.

**Strengths**

1. JOENA combines embedding-based and OT methods, improving network alignment by leveraging the strengths of both approaches.

2. The use of OT theory reduces graph noise, enhancing the robustness and reliability of node alignment.


**Weakness**

1. The method involves complex optimization steps, which may lead to high computational costs, especially for large networks. The paper does not fully address how the model performs with large datasets.

2. Using Random Walk with Restart (RWR) for structure encoding could pose scalability issues for large-scale graphs, as computing RWR scores for all pairs may become infeasible.

3. The model mainly combines existing concepts, such as FGW distance, multi-level ranking loss, and entropy regularized OT. These methods are already well-known, and the model lacks significant novelty.

**Questions:**

How does the proposed method handle scalability when applied to larger and more complex networks?

**Reviewer Confidence:**

3: The reviewer is confident but not certain that the evaluation is correct

**Scope:**

4: The work is relevant to the Web and to the track, and is of broad interest to the community

---

### Official Review · Reviewer_tJwR · 2024-11-30

**Novelty:** 5
**Technical Quality:** 4

**Review:**

The authors propose JOENA, a unified framework for semi-supervised network alignment that integrates embedding-based and OT-based methods. By leveraging OT to enhance embedding learning and embeddings to improve OT optimization, JOENA effectively addresses limitations in existing approaches. Extensive experiments demonstrate significant improvements in both effectiveness (up to 16% MRR improvement) and scalability (up to 20× speedup) over state-of-the-art methods.
________________________________________
Strengths:
- JOENA combines OT and embedding-based approaches in a mutually beneficial way, offering a novel solution for network alignment. The adaptive sampling strategy and learnable OT cost function improve robustness and generalization.
- The paper presents clear theoretical analysis, including insights into embedding collapse and convergence guarantees.
- Comprehensive experiments show superior performance across various datasets and strong robustness to structural and attribute noise. Scalability results demonstrate efficiency, making JOENA practical for large-scale networks.
- The framework offers practical benefits with its learnable cost design and multi-level ranking loss, achieving robust and noise-reduced alignments.
________________________________________
Weaknesses:
- The datasets used do not include highly dynamic or temporal networks, limiting the generalization analysis.
- Memory efficiency is not discussed, which is critical for scaling to extremely large graphs.
- Ablation studies could better isolate the contributions of individual components (e.g., $g_\lambda$, FGW loss).
- The sensitivity of hyperparameters like $\lambda$ is not deeply explored.
- While scalability is demonstrated, the breakdown of runtime contributions from embedding and OT optimization is lacking.
- A more detailed discussion of practical runtime and its dependence on key parameters would be helpful.
________________________________________
Several suggestions for improvements:
- Evaluate JOENA on dynamic or temporal networks to better demonstrate its generalization capabilities.
- Expand ablation studies and provide a more detailed sensitivity analysis of hyperparameters.
- Include a breakdown of runtime contributions and memory usage to validate scalability claims.
- Improve figure clarity and provide a more intuitive visualization of JOENA's workflow.

**Questions:**

Please reply to the weaknesses if I have misunderstandings. Thank you.

**Reviewer Confidence:**

3: The reviewer is confident but not certain that the evaluation is correct

**Scope:**

3: The work is somewhat relevant to the Web and to the track, and is of narrow interest to a sub-community

---

### Official Review · Reviewer_4qBZ · 2024-12-01

**Novelty:** 5
**Technical Quality:** 5

**Review:**

The paper presents a comprehensive exploration of a unified framework JOENA for network alignment, combining embedding-based and OT-based methods. Theoretical analysis, rigorous experimentation, and clear explanations support the claims.

Pos:
1.It is meaningful to incorporate embedding-based and optimal transport-based methods for improved alignment.
2.The paper is well-organized and the figures are clear.

Cons:
1.The paper does not provide sufficient detail on computational resources used for the experiments.
2.The assessment records inference performance only in terms of increasing the number of edges, without considering the impact of growing the number of nodes or changes in network density.

3.More details on the implementation could help reproducibility, such as experiment codes.

**Questions:**

1.What's the training parameter and time of your framework?
2.How sensitive is the method to the choice of the hyperparameter α, which controls the trade-off between Wasserstein and Gromov-Wasserstein distances? Could different α values lead to different alignment results?
3.How feasible is it to implement JOENA in real-world applications with limited computational resources?

**Reviewer Confidence:**

3: The reviewer is confident but not certain that the evaluation is correct

**Scope:**

4: The work is relevant to the Web and to the track, and is of broad interest to the community

---

### Official Review · Reviewer_eBrw · 2024-12-04

**Novelty:** 5
**Technical Quality:** 5

**Review:**

The paper introduces JOENA, a novel framework combining Optimal Transport (OT) with embedding learning for network alignment. JOENA addresses key challenges like robustness to graph noise and scalability, while avoiding embedding collapse. The methodology is well-explained, and the empirical results show significant improvements in alignment quality (MRR) and computational efficiency across multiple datasets.
Pros
1.Innovative Integration: Combines OT with embedding learning, providing a novel solution to network alignment challenges.
2.Robustness: Demonstrated robustness to both structural and attribute noise.
3.Scalability: Efficient at handling larger graphs compared to other OT-based methods.
4.Strong Theoretical Foundation: Clear explanation of the convergence properties and handling of embedding collapse.
Cons
1.Large-Scale Networks: The paper does not fully explore how JOENA performs on extremely large networks with millions of nodes and edges, particularly in real-time or dynamic scenarios.
2.Hyperparameter Sensitivity: The method depends on hyperparameters like λ, and more guidance on tuning for specific tasks would be useful.

**Questions:**

1.Sensitivity to Parameters and Missing Data:
While JOENA benefits from an adaptive sampling strategy, it relies on the parameter λ for effective learning. How sensitive is the model to variations in λ, and how can this parameter be reliably selected in practice? Additionally, how does JOENA handle scenarios with severe missing data (e.g., large portions of edges or attributes missing)? Can the model maintain robustness under such challenging conditions?

2.Performance on Large and Dynamic Networks:
The experiments demonstrate scalability, but how does JOENA perform on extremely large-scale networks with millions of nodes and edges, or in dynamic network scenarios where the structure evolves over time? Would significant modifications be required to adapt the method to such real-time applications?

**Reviewer Confidence:**

3: The reviewer is confident but not certain that the evaluation is correct

**Scope:**

4: The work is relevant to the Web and to the track, and is of broad interest to the community